# Interconnected Challenges: Examining the Impact of Poverty, Disability, and Mental Health on Refugees and Host Communities in Northern Mozambique

**DOI:** 10.3390/healthcare13243187

**Published:** 2025-12-05

**Authors:** Theresa Beltramo, Florence Nimoh, Sandra Sequeira, Peter Ventevogel

**Affiliations:** 1United Nations High Commissioner for Refugees, CH-1201 Geneva, Switzerland; ventevog@unhcr.org; 2School of Economics and Management, University of Geneva, CH-1211 Geneva, Switzerland; 3World Bank Group, Nairobi P.O. Box 30577-00100, Kenya; florencenana.nimoh@gmail.com; 4Department of International Development, London School of Economics and Political Science, London WC2A 2AE, UK; s.sequeira@lse.ac.uk

**Keywords:** refugees, mental health, anxiety, depression, host community members, integration, forced displacement, conflict, disability, poverty, Mozambique

## Abstract

**Background:** Poverty, disability, and mental health may reinforce one another. Forced displacement can compound these challenges, yet comparable data on displaced and non-displaced groups in the same setting are scarce. This study examines associations among mental health, disability, pessimism, loneliness, self-esteem, and financial security for refugees and nearby host communities in Mozambique. **Methods:** Ultra-poor adults—refugees (n = 134) and Mozambican nationals living near the settlement (n = 314)—were identified using a World Bank poverty scorecard. Surveys captured depression (PHQ-9), anxiety (GAD-7), disability (Washington Group Short Set), and socioeconomic characteristics. **Results:** Symptom rates are high in both groups—depression: 34% (refugees) vs. 29% (hosts); anxiety: 25% overall—with women reporting higher levels. Disability prevalence is substantial (refugees 25%; hosts 22%). Respondents with disabilities show markedly higher rates of depression (≈2×) and anxiety (≈3×). Financial security is negatively associated with symptom scores: a one-unit-higher financial security index correlates with a 0.069 lower anxiety score (*p* < 0.05) and a 0.069 lower depression score (*p* < 0.01). Pessimism is positively associated with poorer mental health; anxiety and depression are more than 2.5× as prevalent among chronically pessimistic respondents. Loneliness shows no clear association with anxiety or depression in this sample, whereas low self-esteem is strongly associated with both; prevalence of GAD and depression is more than twice as high among those with low self-esteem. **Conclusions:** We document strong associations between poverty, disability, and mental health. These patterns underscore the importance of strengthening mental and public health services for both refugees and hosts, with particular attention to women and disabled individuals.

## 1. Introduction

Depression and anxiety disorders are among the most common mental health illnesses: around 3–4% of the world’s population suffers from each at any given time [1] and together they are responsible for 8% of years lived with disability globally [2]. Mental disorders can have significant negative impacts on socioeconomic outcomes, such as employment and productivity [3,4]. These impacts are likely even stronger in fragile and conflict-affected parts of the world, particularly among populations exposed to collective violence and forced displacement. Conflict and displacement can adversely impact mental health and psychosocial well-being [5,6]. Exposure to stressful and potentially traumatic events and multiple interpersonal losses greatly increases the risk of developing mental disorders such as depression, anxiety disorders, and post-traumatic stress [7]. Increased rates of mental health conditions among refugees are also fuelled by post-forced displacement conditions such as poor socioeconomic conditions, lack of basic rights (such as the rights to work, open bank accounts, register for a mobile phone), lack of legal documentation, difficulty in getting jobs or accessing higher education, as well as separation from family and supportive community [2,8,9,10].

One in five persons living in conflict affected areas are estimated to suffer from mental health conditions, such as depression, anxiety, and/or psychosis, and one in ten displaced individuals are living with moderate or severe forms of such mental health conditions [7]. Recent evidence indicates that depression rates among refugees rose more significantly than those of host populations during the COVID-19 pandemic and that depression rates are generally significantly higher for refugees than those of host populations [11,12]. Surveys in Uganda found that 50% of refugees reported experiencing depression symptoms during the pandemic vs. 5% of nationals [8]. This highlights the urgent need for better tracking of mental health statistics among displaced populations and hosts, especially in times of shocks [7].

A large body of research links poverty and mental health: those with the lowest incomes are typically 1.5 to 3 times more likely to experience depression or anxiety compared to those who have higher incomes [13,14]. Rates of depression, anxiety, and suicide correlate negatively with income [13,14,15,16] and employment [17,18]. Although comparable evidence on this relationship among refugees and their host communities remains scarce, the available evidence shows that refugee populations generally experience higher poverty rates than nationals, regardless of the policy environment [19]. This can be especially salient where refugees lack freedom of movement and are relegated to camps. In such situations, refugees tend to be concentrated in poorer areas where locals are disproportionately poor compared to the national average.

Recent research on poverty within refugee households highlights substantial heterogeneity that warrants further study. For example, evidence from Jordan shows lower poverty rates among refugees living in camps (about 42%) than among those residing in host communities outside these settlements (about 62%) [20]. Moreover, recent evidence indicates that, within households in Kenya and Uganda, poverty is not equally distributed among all members. Refugee children are up to three times more likely to experience poverty compared to adults [21]. This evidence highlights the need for further research to understand the diverse nature of poverty among refugee groups, which will aid in effectively targeting assistance to the most vulnerable.

Having a disability is a high-risk factor for developing mental health conditions, including depression or anxiety [22,23]. According to the WHO, disability arises from the interaction between health conditions and various personal and environmental factors, such as negative attitudes, inaccessible transportation and public buildings, and limited social support. The existing literature also highlights that individuals with physical disabilities have higher rates of depression and poor mental health outcomes, and this is particularly the case among women [24,25]. A U.S. study finds that adults with disabilities report experiencing mental distress almost five times as often as adults without disabilities [26]. There is limited evidence on the incidence of disability among refugees and its correlation with mental health and poverty. Additionally, comparative data on disability between refugees and host communities in the same setting is scarce. The evidence that does exist suggests that refugees tend to have higher burdens of disabilities than host communities [27,28] and can face more difficulties in accessing services due to their limited rights in the country of asylum and/or relative levels of poverty (and diminished ability to pay for health services) compared to host community members.

In this paper, we document links between poverty, disability and mental health for both ultra-poor refugees and host community members. We also explore correlations across different measures including loneliness and pessimistic life views to see how they correlate with depression, anxiety, disability, and financial security. Loneliness has been characterized as an epidemic in modern society, impacting one-third of individuals in developed countries [29]. It is defined as an ‘unpleasant experience that arises when a person’s social network is deficient in some significant way, either quantitatively or qualitatively’ [30]. Loneliness tends to be more prevalent and intense among individuals with severe mental illness [31]. Personality has been identified as a significant predictor of depressive symptoms [32,33,34]. In particular, dispositional optimism and pessimism are noted as key predictors of depression risk in numerous studies [35,36,37,38], as these traits are closely linked to the absence of positive emotions, a core symptom of depression [39]. Limited work to date has evaluated additional measures associated with mental health, such as loneliness and pessimistic life views, in low-income settings.

Understanding the levels of mental health outcomes among poor refugees and host communities and how mental health relates to poverty and disability is critical in informing evidence-based (and differentiated) programming as needed. This study looks at a forced displacement context for impoverished refugees and hosts in Mozambique and is one of the first studies to assess how loneliness and pessimism bias correlates with mental health and disability outcomes. We examine associations across measures of mental health, disability, loneliness, pessimism bias, and financial security (a proxy for welfare) among the poorest segments of both refugees and host communities in a forced displaced setting in Mozambique, providing insights into the compounded challenges faced by the poorest segments of forcibly displaced populations, which have been underrepresented in the existing literature.

### Poverty and Displacement in Mozambique

The World Bank estimates that 63% of the population in Mozambique is living in poverty. Mozambique’s protracted civil war (1977−1992) forced more than a third of the population at the time—more than four million civilians of the then 12 million total population—to flee to the countryside, to cities, and to refugee camps and settlements in neighbouring countries, leaving long-lasting socioeconomic scars. A recent study revealed that individuals displaced to cities, smaller towns, or rural settlements invested significantly more in education compared to children who stayed in the countryside or who were displaced to refugee camps in neighbouring countries. However, those who were displaced exhibited significantly lower social capital and worse mental health, even three decades after the end of the war. These findings suggest that displacement shocks in Mozambique triggered positive human capital investments, breaking links with subsistence agriculture, but also causing long-lasting, social, and psychological consequences [40].

Mozambique hosts over 24,000 refugees as of July 2025 [41]. Long asylum procedures and generalized rural poverty have posed formidable obstacles to socioeconomic integration. The Maratane refugee settlement is situated in one of the most populous provinces in Mozambique. The site has evolved from a traditional refugee camp into a more open settlement, reflecting the Mozambican government’s policy shift towards promoting local integration and sustainable solutions. The Maratane Refugee Settlement is home to about 13,190 refugees and asylum seekers, primarily from the Democratic Republic of Congo, Burundi, Rwanda, and Somalia. About 16,390 Mozambican nationals from the surrounding community directly depend on the settlement for services (e.g., health, education, etc.) provided by the United Nations High Commissioner for Refugees (UNHCR), NGO partners, and the government. Many Mozambican nationals (99%) living near the settlement are Makua (or Makhua), a Bantu-speaking ethnic group. The Makua live in an extensive area that covers the northern parts of Mozambique and includes the Cabo Delgado, Niassa, Zambézia, and Nampula provinces. Their traditional livelihoods are based on small-scale subsistence farming.

## 2. Methods and Materials

### 2.1. Aims

We obtained measures of depression and anxiety, pessimistic life views and loneliness, socioeconomic outcomes, and disability for both refugees and host communities in a low-income setting, and among the poorest segments of the population. This allowed us to help build an evidence base on comparable mental health and socioeconomic outcomes for ultra-poor refugees and hosts.

### 2.2. Survey Data

The data are drawn from a set of surveys covering a range of thematic areas such as housing characteristics, employment and livelihood, education, finance and assets, health, social capital, social norms, consumption expenditure, and food security. These surveys were part of a wider impact evaluation of the Graduation Programme [42] targeting poor refugees and host community members living in and around Maratane Settlement [43]. The graduation programme in Maratane includes assistance in the following areas, with the aim of providing encouragement and improving self-esteem, as well as providing personalized interventions for individuals’ specific needs: consumption support, instructions on resume writing, core skills training, language and financial literacy classes, market-oriented skills and vocational training, job support, asset transfer, and coaching services. The programme also includes support for self or wage employment including a paid apprenticeship to improve linkages to jobs so that participants with limited experience could increase their employability. All measures included in this study were taken prior to the main intervention associated with a large cash transfer and employment support that was provided in August 2021 [43].

The selection of ultra-poor households was designed based on a poverty score card employed by the World Bank for the targeting of Government social protection programmes. This process followed five key steps summarized in Figure 1, namely the following: (1) pre-screening of eligible refugees to identify those who were not engaged in regular employment; (2) verification of consent to participate in the programme; (3) collection of socioeconomic data through a poverty scorecard survey, which contained basic questions about their family, language skills, housing, and household assets; (4) calculation of poverty score and poverty screening; (5) random selection (50%) of control/treatment groups. Refugees and Mozambicans who reported an interest in participating in the programme and met the basic pre-screening criteria were administered a questionnaire that used 10 indicators to identify ultra-poor households. The poverty scorecard was based on the Government of Mozambique’s Simple Poverty Scorecard [44]. Annex 1 provides more details on the questionnaire and scoring approach. In total, 448 individuals (18 years and older) were selected with 134 being refugees and 314 belonging to the Mozambican host community. In addition, of the 448 some 166 were randomly selected to participate in the graduation programme, while 282 remained in the control group. Because the survey wave which we exploited for this paper took place prior to the main intervention, we utilized the sample of 448 responses to generate the descriptive results outlined in this paper.

### 2.3. Enumerator Training and Procedure

Two surveys were employed for this analysis. The baseline survey was conducted from September–December 2019, while the first follow-up survey—Wave 2—took place from May to July 2021. Both surveys took place in person and prior to the main component of the project—the large, unconditional cash intervention.

The questionnaire was designed in both English and Portuguese. Only 44% of refugees and 52% of nationals spoke Portuguese, Mozambique’s national language. The primary languages spoken by refugees include Swahili (96%) and French (53%), while nationals spoke primarily Emakua (99%). For respondents who did not understand English or Portuguese, the questions were asked in Emakua or Swahili by the enumerators during the interview.

The survey was administered by six enumerators who were fluent in Portuguese and either Emakua (three) or Swahili (three). Enumerators were selected based on their proficiency in Emakua or Swahili, and on their experience in data collection. All enumerator candidates were subject to a rigorous in-person language proficiency test administered by official translators in each language hired by the project team. Before the start of the data collection, the enumerators underwent an intensive three-day training to ensure that the questionnaire was well-understood and correctly administered. The training focussed on the required competencies for enumerators using role-play scenarios and going through the interview questionnaires, ensuring that enumerators understood key concepts in both Portuguese and the local languages. The interviews were done via Computer Assisted Personal Interviewing (CAPI).

### 2.4. Measures

**Depression:** Symptoms of depression were measured with the Patient Health Questionnaire (PHQ-9), a common self-reported measure for depressive symptoms [45]. Its nine items ask how often the respondent has been bothered over the last two weeks: (i) little interest or pleasure in doing things; (ii) feeling down, depressed or hopeless; (iii) trouble falling asleep, staying asleep or sleeping too much; (iv) feeling tired or having little energy; (v) poor appetite or overeating; (vi) feeling bad about yourself—or that you are a failure or have let yourself or your family down; (vii) trouble concentrating on things, such as reading the newspaper or watching television; (viii) moving or speaking so slowly that other people could have noticed; or, the opposite—being so fidgety or restless that you have been moving around a lot more than usual; and (ix) thoughts that you would be better off dead or of hurting yourself in some way. The overall score can range from 0 to 27, with scores of 0–4 indicating no depression, 5–9 = mild depression, 10–14 = moderate depression, 15–19 = moderately severe depression and ≥20 = severe depression. The PHQ-9 has been validated for use in Mozambique [46] and has been used in research with Mozambican populations [47,48].

**Anxiety:** Symptoms of anxiety were measured using the Generalized Anxiety Disorder 7-item (GAD-7), a standardized tool screening for anxiety symptoms, in particular Generalized Anxiety Disorder. The seven questions explore how often over the last two weeks the respondent has been bothered by the following: (i) feeling nervous, anxious or on edge; (ii) not being able to stop or control worrying; (iii) worrying too much about different things; (iv) trouble relaxing; (v) being so restless that it is hard to sit still; (vi) becoming easily annoyed or irritable; (vii) feeling afraid as if something awful might happen. The GAD-7 score is calculated by assigning scores of 0, 1, 2, and 3, to the response categories of “not at all,” “several days,” “more than half the days,” and “nearly every day,”, respectively. The total score ranges from 0 to 21. The tool has been validated for use in Mozambican populations [49] and has been used in research in Mozambique [47,48] and among Swahili-speaking populations in the region [50,51]. For this research, we used a score of 10 as cut-off point for a likely anxiety disorder [52].

**Self-esteem:** To measure one’s self-worth, we used seven questions from the ten-item Rosenberg Self-Esteem scale [53]: (i) I am able to do things as well as most other people; (ii) I feel useless at times; (iii) I feel that I am a person of worth, at least on an equal plane with others; (iv) I feel that I have a number of good qualities; (v) I feel I do not have much to be proud of; (vi) I take a positive attitude toward myself, and (vii) I am satisfied with my life. This scale measures both positive and negative feelings about oneself, and questions are answered using a 4-point scale ranging from strongly agree to strongly disagree. Questions that ask about negative feeling about oneself are reversely scored. Scores from each question are then aggregated, with a higher score indicating a higher self-esteem. The Rosenberg Self-Esteem Scale has been very widely used including in Sub-Saharan Africa including in Mozambique and among Swahili-speaking populations. We categorize low self-esteem as those individuals falling in the lowest quartile of the scale.

**Loneliness:** To measure subjective feelings of loneliness, four questions were asked that were drawn from the UCLA loneliness scale [30]: (i) How often do you feel that there is no one you can turn to? (ii) How often do you feel isolated from others? (iii) How often do you feel that your interests and ideas are not shared by those around you? (iv) How often do you feel you have a lot in common with the people around you? Questions that ask about negative feelings about loneliness are reversely scored. Scores from each question are then aggregated, with a higher score indicating a higher sense of loneliness. Respondents rated each question on a scale of 0 (Never) to 3 (Often), based on which we generate an adapted total loneliness score from the four questions by summing the numerical values assigned to each of the responses.

**Pessimism:** We measured value-based questions reflecting pessimism by asking respondents how they rank their life at the time the survey was implemented and how they expect it to be in five years’ time, with 0 being the worst and 10 being the best. Individuals are ranked as being “chronically pessimistic” if they rank their life as a 3 or below both now and in five years’ time.

**Disability:** We assessed self-reported disability using the Washington Group on Disability Statistics-Short Set on Functioning (WG-SS) consisting of six questions, measuring the difficulties a person may have in undertaking basic functioning activities including seeing, hearing, walking or climbing stairs, remembering or concentrating, self-care, and communication (expressive and receptive). Respondents are considered as having a disability if they report ‘having a lot of difficulties’ or ‘cannot do at all’ in at least one of the six questions. The tool is widely used in population surveys including in humanitarian settings [54].

**Financial security:** To measure financial security we generated a Financial Security Index which is an index of four proxies for financial security (ease of paying a surprise bill, percent of income saved last month, take home monthly pay, and whether the respondent is engaged in casual employment), constructed by first equally weighting the average z-scores of each indicator that composes each dimension and then averages across the four measures once standardized.

### 2.5. Data Collection and Statistical Analysis

All data were collected using a Computer Assisted Personal Interviewing (CAPI) through ODK, an open-source mobile data collection platform. Plausibility checks were performed, and data completeness was assured prior to analysis. We present the means for each score for refugees and Mozambicans, and for men and women within each population group. We use *t*-tests to assess equality of means between the groups and report the associated *p*-values. To determine whether the refugees and the host community members with significant depression or anxiety symptoms are more or less financially secure than those without these conditions, we employed an Ordinary Least Squares (OLS) regression analysis. We regress the likelihood of suffering from Generalized Anxiety Disorder (GAD) or depression on the financial security index, incorporating sociodemographic control variables such as age, sex, years of education, household size, housing type, Portuguese fluency, and refugee status.

## 3. Results

### 3.1. Description of Demographics

Table 1 shows the characteristics of refugees and Mozambicans measured at baseline. Notably there are statistically significant differences across the refugee and Mozambican communities including that refugees are slightly older, with an average age of 39 vs. 36 years for hosts. Refugees also have higher levels of schooling compared to locals, while Mozambicans are more likely to be married—76% vs. 47%.

### 3.2. Anxiety

We find similar rates of anxiety for both refugees and hosts: one out of four respondents are likely to experience anxiety symptoms. This figure is slightly higher among women than men, but these differences are not statistically significant (Figure 2). Considering how anxiety varies across age groups, among refugees and hosts, heads of household who are above 54 years are slightly more likely to suffer from anxiety disorder compared to younger heads of household though results are not statistically significant.

GAD can cause serious distress and interfere with one’s daily functioning such as work, school, social activities, and relationships. When asked “how difficult have these problems (related to anxiety) made it for you to do your work, take care of things at home or get along with other people?”, 74% of refugees and 52% of Mozambicans who are likely to experience GAD cite this is somewhat to extremely difficult (Figure 3). When asked “how difficult have these problems (related to anxiety) made it for you to get along with other people?”, 59% of refugees and 48% of Mozambicans report ‘very difficult to extremely difficult’, highlighting the possibility of anxiety underpinning social tension (Figure 4). There are no statistically significant differences across Mozambicans and refugees in the outcomes displayed in Figure 3 and Figure 4.

### 3.3. Depression

Some 34 percent of refugees in the Maratane settlement report experiencing moderate or severe depression disorder, compared to 29 percent of hosts (Figure 5). A non-parametric test suggests that these differences are significant, with refugees exhibiting slightly higher prevalence of symptoms of depression.

Every 28 out of 100 refugee women suffer from moderately severe (19) or severe depression (9), compared to 14 out of 100 refugee men (*p* < 0.04). We further find that 53 percent of refugees and 46 percent of Mozambicans suffer from mild depressive episodes.

### 3.4. Self-Esteem

Mean self-esteem scores are identical for refugees and hosts (both 21) (Figure 6). There are no statistical differences between men and women. The bottom quartile has a score from 14 to 19 and the upper quartile has a score from 23 to 27.

We find that self-esteem is associated with depression and anxiety. Having low self-esteem (bottom quartile of the index) more than doubles the likelihood of the prevalence of Generalized Anxiety Disorder and depression. Similarly, high self-esteem (upper quartile of the index) is associated with lower likelihood Generalized Anxiety Disorder and depression by a factor of 8 (Table 2).

### 3.5. Loneliness

Both refugees and hosts report similar levels of loneliness—5.6 for Mozambicans and 5.5 out of 10 for refugees (Figure 7). There are no statistical differences between men and women. The bottom quartile has a score between 1–4.5 and the upper quartile between 7–10.

We find that loneliness is not strongly associated with our measures of Generalized Anxiety Disorder or depression (Table 3).

### 3.6. Pessimism Bias

We assessed whether pessimistic perceptions of one’s life are correlated with anxiety or depression in the community studied. Some 78% of refugees and 75% of Mozambicans believe their life is a 3 out of 10 or below at present. Some 28% of refugees and 34% of Mozambicans believe that, in 5 years, their life will be a 3 out of 10 or below.

For individuals identified as ‘chronically pessimistic,’ the prevalence of Generalized Anxiety Disorder is 82% among refugees and 67% among Mozambicans (*p* < 0.10). The prevalence of depression is 82% for refugees and 77% for Mozambicans (*p* < 0.00), which is more than 2.5 times the population rate (Figure 8 and Figure 9).

### 3.7. Financial Security

To examine whether refugees and host community members who are depressed or anxious are more or less financially secure compared to those who are not, we regressed the data for those who are likely to suffer from Generalized Anxiety Disorder (depression) on the financial security index alongside sociodemographic control variables (including age, sex, years of education, household size, housing type, whether the individual reports fluency in Portuguese, and whether they are a refugee).

Table 4 shows the findings for the pooled sample, as well as for the individual samples of refugees and host community members. The pooled sample of refugees and hosts who are classified as anxious are also less financially secure at baseline. This result is statistically significant at the 5% level. Similarly, refugees and hosts who are classified as depressed are also relatively more financially insecure at baseline, a correlation coefficient that is significant at the 1% level. We find financial security is inversely associated with mental health issues: a one-unit increase in financial security is associated with a 0.069 unit decrease in anxiety (*p* < 5%) and a 0.069 unit decrease in depression (*p* < 1%). Consistent with our findings above, refugees are more likely to be depressed than hosts and women are more likely overall to be depressed, as are individuals with less education. Notably the results are driven by the host community. Among refugees, the financial security coefficient is negative for both depression and anxiety, but neither estimate is statistically significant.

### 3.8. Disability

Both refugees and hosts report high levels of disabilities compared to global averages, and suffering from disabilities is associated with having symptoms of depression or anxiety. In our sample, 25% of refugees and 22% of Mozambicans report having a disability as measured by the Washington Group on Disability Statistics—Short Set on Functioning (WG-SS). Refugees and Mozambicans with disabilities are more than twice as likely to suffer from depression and three times as likely to suffer from anxiety compared to respondents without disabilities.

## 4. Discussion

Our results suggest that levels of depression, anxiety, loneliness, pessimism, and disability are high among refugees and host Mozambicans, though refugees have a higher likelihood of experiencing moderate or severe depression, especially women. The prevalence of depression and anxiety disorder in our sample is considerably higher than estimates for African countries [55]. Recent studies in Mozambique found a prevalence of depression with the PHQ-9 of 10% among women and 2% among men, and a prevalence of anxiety disorder (measured with the GAD-7) of 11% among women and 2% among men [56]. The estimates in our sample are also higher than those of Audet et al. (2018) who found a 14% prevalence of depression (as measured with the PHQ-8) among women-headed households in rural Mozambique [57]. Consistent with global evidence, we find that those who suffer from depressive symptoms also experience anxiety symptoms. In our sample, eight out of ten persons with moderate or severe depressive disorder are also likely to be suffering from anxiety disorder, with the figures being slightly higher among women.

A growing body of research has established links between exposure to conflict and traumatic events and poor mental health outcomes. Refugees are disproportionally exposed to various stress factors which can negatively affect their mental health and well-being during the time of forced displacement including exposure to armed conflict, violence, trauma, and persecution, as well as after becoming refugees [7]. Our findings suggest, however, that ultra-poor host community members are suffering nearly as high rates of depression, the same level of anxiety, similar levels of loneliness, and only moderately lower levels of pessimism in comparison to refugees. One possible factor that contributes to the high prevalence of depression among impoverished Mozambicans is the long-term impact of trauma from the Mozambican civil war (1977–1992), compounded by ongoing poverty. There is growing evidence that the effects of exposure to violence and forced displacement can be long-lasting [19,40]. It is also possible that host community members feel detached from the broader Mozambican community. Very few reported having friends or knowing anyone in the nearest city and less than 50% speak the official language of Portuguese. Finally, proximity to refugees who are often perceived as having privileged access to humanitarian aid may further deteriorate the mental health of host community members. If left unaddressed, high mental health burdens among both refugees and hosts may hinder refugees’ social and economic integration—an element widely seen as critical for broader economic stability and the pursuit of durable solutions for refugees.

Women have unequal mental health burdens. We find that women in the poor refugee communities in Mozambique are twice as likely to suffer from moderately severe or severe depression than men. This is in line with previous research that suggests that women and girls in conflict settings face biological and socioeconomic factors that make them more vulnerable to mental health disorders and has also been found in nationwide prevalence study in Mozambique [56]. Further, women can tend to suffer from lack of agency prevalent in most societies due to patriarchal tendencies which limit women’s choices and can lead to worse mental health outcomes.

We further examine the role of pessimism, subjective feelings of loneliness, and self-esteem on depression and anxiety rates in our sample. Among our sample, ‘Chronic Pessimists’ have a prevalence of Generalized Anxiety Disorder that is more than 3 times higher for refugees and more than 2.5 times higher for hosts. Additionally, the prevalence of depression is more than 2.5 times higher for both groups. This is consistent with evidence that finds patients suffering from Major Depressive Disorder tend to be more pessimistic in general [58]. Although evidence shows that asylum seekers and refugees frequently experience loneliness and isolation due to factors like language barriers, poverty, and lack of social support, among others [59], loneliness is not associated with depression and anxiety in our sample. The lack of association between loneliness and depression/anxiety should be interpreted cautiously. First, using a four-item version of the UCLA scale likely increases measurement error, biasing estimates toward zero. Second, in this setting social strain may manifest through obligations, crowding, or network dependence rather than perceived isolation, which our items may capture imperfectly.

Self-esteem, or lack thereof, is a positive predictor of depression and anxiety for both refugees and hosts, doubling the likelihood of these conditions. Evidence to date confirms that low self-esteem and depression and anxiety are strongly related, though more research is needed to learn on the causality of the relationship—i.e., whether depression causes poor self-esteem, or vice versa. A meta review of the literature indicates that self-esteem has a significantly stronger impact on depression than depression has on self-esteem. In contrast, the effects of self-esteem and anxiety on each other are relatively balanced [60]. This result suggests that in our refugee and host communities, mental health services would be more effective if they were integrated with programs aimed at improving self-esteem.

Being financially secure is associated with a lower likelihood of experiencing both anxiety and depression symptoms. This is consistent with recent research that has established a bidirectional causal relationship between poverty and mental illness [13]. These findings raise important questions about the drivers of poor mental health outcomes across refugee and host communities. There may be a case for treating mental health interventions as a complementary component of poverty reduction efforts, in light of their potential economic payoffs.

The negative association between depression or anxiety and financial insecurity was driven by the host community, while the association between depression and anxiety for refugees was not statistically significant. This could be due in part to the relatively small sample size of our refugee population, which limits the power to detect a relationship. On the other hand, our refugee and host categories can potentially also mask within-group heterogeneity. Among refugees, national-origin subgroups differ in exposure to trauma, time since displacement, language, and network structure, all of which may shape mental health and its relationship to economic stress. In our data, refugees also have higher average schooling than hosts. Education, stronger enclave networks, differential access to NGO support, or stigma/reporting norms could plausibly buffer the mental health impact of short-run financial insecurity, yielding smaller observable associations even absent true differences. Given sample size constraints, we are underpowered to test these mechanisms or to present credible subgroup estimates, and we avoid over-interpreting pooled contrasts. We suggest targeted adequately powered heterogeneity analyses as a priority for future work.

Another important conclusion from the link between financial security and reduced mental health burdens is that, for both impoverished refugees and Mozambicans, mental health and poverty appear to be interlinked. This finding is crucial for policymakers suggesting that in Mozambique (and potentially other contexts) poor host community members should be included in development programming aimed at improving mental health and reducing poverty.

Both refugees and Mozambicans with disabilities are more than three times as likely to suffer from anxiety and more than twice as likely to suffer from depression compared to those without disabilities. The disproportionate prevalence of anxiety and depression among persons with disabilities underscores the need for policy action to improve access to health services for both refugee and host communities in Mozambique. This would help to reduce the overall burden of treatable disabilities, such as eye conditions like trachoma and cataract [61] and enhance the overall health of individuals living with disabilities. Additionally, employment programs should prioritize the inclusion of individuals living with disabilities to improve mental health among both refugees and hosts.

The findings contribute to the academic discourse by highlighting the intricate relationships between these critical factors, thereby informing more effective policy interventions and support mechanisms for vulnerable populations in similar settings. Understanding the levels of mental health outcomes among poor refugees and host communities and how mental health relates to poverty and disability is critical to inform evidence-based (and differentiated) programming as needed.

### Limitations

Given limitations of interview time we had to shorten the full scale of the UCLA Loneliness scale Survey. In future assessments it would be useful to administer this full scale as well as the Harvard Trauma Questionnaire and particularly the Adverse Childhood Experiences Scale to understand more fully the relationship between daily life stressors, trauma, and adverse childhood experiences and mental health and disability outcomes. Further, the chronically pessimistic scale is the author’s invention and the cut-off of a score of a 3 out of 10 for both today and in five years would benefit from more robust experimentation to validate the cut-offs selected.

Our sample is also restricted to ultra-poor refugees and host community members. Further future work is needed on this topic to better understand a more granular outcome for different segments of poor refugee and host community members and map heterogeneity of poverty among particularly refugee communities. We are limited by our sample size in our ability to undertake analysis based on age groups as well. As a result, we cannot disaggregate findings for adolescent mental health outcomes, nor different household configurations such as women-headed households. These questions are important to be able to target limited resources effectively.

While the Patient Health Questionnaire 9 (PHQ-9) and the Generalized Anxiety Disorder 7 (GAD-7) questionnaires are standard questionnaires used in Mozambique and many global contexts, the measures of probability to be depressed or anxious that are produced are likely to be noisy compared to the gold-standard clinical interviews diagnosing depression and anxiety. Due to logistical and ethical constraints, we relied on these self-reported measures. Future studies should make greater efforts to validate the PHQ-9 and GAD-7 questionnaires against clinical psychologist interviews. This will further our understanding of the correlation between the measures and the relative noise of the PHQ-9 and GAD-7. Finally, we also note a further limitation that, while the enumerators were bilingual and extensively trained, we did not deploy formally translated and back-translated Emakua/Swahili instruments; this may have introduced measurement error and cultural misalignment in the mental health scales. It is recommended that future surveys translate into all local languages to minimize any error in translation.

We also note that the small sample size limits statistical power, constraining our ability to explore subgroup heterogeneity in greater depth and to robustly evaluate the causal relationship between mental health outcomes and welfare. Future research is needed with larger, more representative samples and longitudinal designs to enable stronger causal inference and a more comprehensive understanding of heterogeneity across key subpopulations.

## 5. Conclusions

Our findings document a link between poverty and mental health, introducing disability as another contributing factor among poor refugees and host communities in Mozambique. Both refugees and host Mozambicans experience high levels of depression, anxiety, loneliness, pessimism, and disability with refugees, especially women, being more likely to suffer from moderate or severe depression. Notably, the incidence of depression and anxiety among these groups is significantly higher than the global average in Mozambique and in Africa, more generally. Our results reinforce that exposure to conflict and displacement and ongoing poverty are major contributors to mental health among both refugees and host communities. This underscores the need for enhanced mental health and public health services for both refugees and hosts, with a particular focus on women.

Pessimism and low self-esteem are strongly associated with depression and anxiety. Mental health services should integrate programmes aimed at improving self-esteem. Further, our results demonstrates that financial security is strongly associated with reduced risks of anxiety and depression among refugees and host communities. This underscores the need for integrated approaches that combine economic and psychosocial support in forcibly displaced settings in Mozambique. Livelihoods interventions should therefore incorporate mental health components to ensure that poverty alleviation strategies also address psychological well-being. As a direct policy outcome stemming from this research, UNHCR Mozambique has decided to require all 2026 funded programmes in Maratane Settlement for refugees and hosts related to livelihoods to include an explicit mental health component. In addition, standardized mental health measures (PHQ-9 and GAD-7) will be integrated into UNHCR protection monitoring tools to enable consistent and systematic tracking of psychological well-being.

Persons with disabilities are significantly more likely to suffer from anxiety and depression, highlighting the need for improved access to health services and inclusive employment programmes. Additionally, poor mental health outcomes can hinder the economic and social integration of refugees, which is essential for the economic growth and stability needed for long-term solutions for refugees. Development programmes should include both refugees and poor community members to effectively address mental health and poverty.

To effectively address the interconnected challenges of poverty, disability, and poor mental health in low-resource refugee settings, integrated interventions are essential. These should combine psychosocial support with inclusive livelihood opportunities. For instance, community-based mental health initiatives can be linked with vocational training, cash-for-work programs, or microenterprise support tailored to individuals with disabilities and those experiencing depression or anxiety—particularly women, who are disproportionately affected. Additional robust, context-specific research is needed to identify the most impactful combinations of mental health and livelihood interventions for both refugees and host communities, to guide evidence-based programming.

This paper is the first to document a relationship across measures of mental health, disability burden, pessimism, subjective feelings of loneliness and self-esteem, and socioeconomic factors, including financial security, among the poorest segments of both refugee and host communities in in a forced displaced setting in Mozambique. We find comparably high levels of symptoms of depression, anxiety, loneliness, pessimism, and disability across both poor refugees and surrounding host communities in Northern Mozambique.

Our findings also confirm the relationship between poor mental health and poverty, aligning with recent research that has established a directional relationship between the two. A recent systematic review and meta-analysis showed that psychosocial interventions for common mental disorders, such as depression and anxiety, improve labour market outcomes, including employment, time spent working, capacity to work, and engagement in job search [62]. Despite growing evidence, few livelihoods or employment programmes are paired with interventions that address depression and anxiety—particularly for forcibly displaced populations. More empirical research is needed on inclusive programmes that jointly target poverty and/or employment and mental health, and on their effects for both displaced people and the communities that host them.

## Figures and Tables

**Figure 1 healthcare-13-03187-f001:**
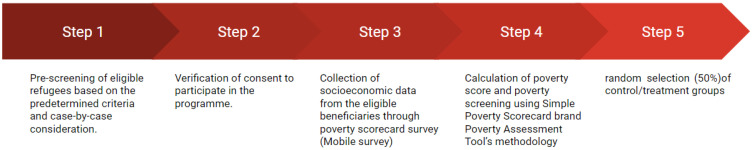
Overview of beneficiary selection pre-screening and poverty verification.

**Figure 2 healthcare-13-03187-f002:**
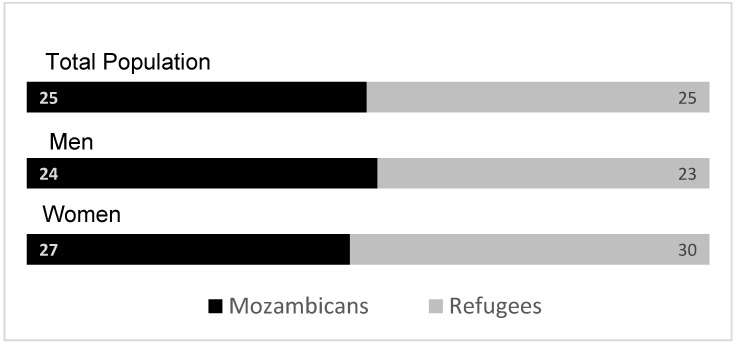
Prevalence of symptoms of anxiety disorder by gender. Source: Mozambique Impact Monitoring First Follow-up Survey (2021); Notes: A non-parametric Wilcoxon rank-sum test of the null hypothesis that for randomly selected values of each population (refugees and hosts); the probability of the value for refugees being greater than the value for Mozambicans is equal to the probability of the value for Mozambicans being greater than the value for refugees yields the following *p*-values: all comparison between refugees and Mozambicans (*p* = 1); refugee men vs. Mozambican men (*p* = 0.96) and refugee female vs. Mozambican female (*p* = 0.80).

**Figure 3 healthcare-13-03187-f003:**
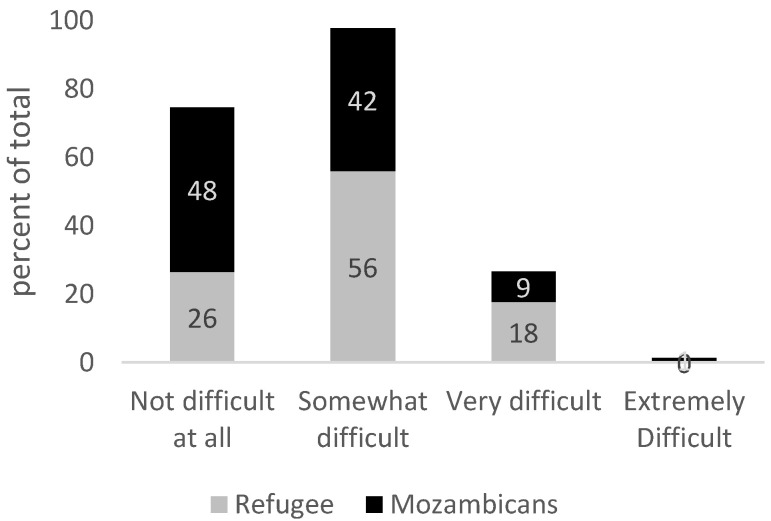
Responses to the following question: “How difficult have these problems (related to anxiety) made it for you to do your work, take care of things at home or get along with other people?” Source: Mozambique Impact Monitoring First Follow-up Survey (2021). Notes: A non-parametric Wilcoxon rank-sum test of the null hypothesis that for randomly selected values of each population (refugees and hosts), the probability of the value for refugees being greater than the value for hosts is equal to the probability of the value for hosts being greater than the value for refugees yields the following *p*-values: *p* = 0.71.

**Figure 4 healthcare-13-03187-f004:**
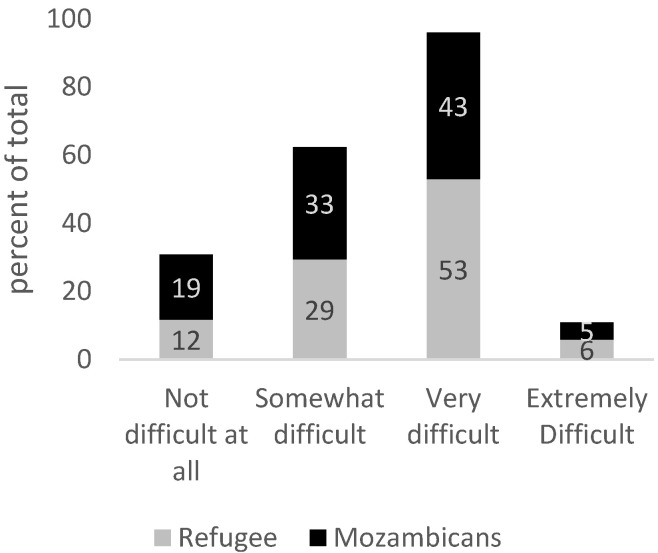
Responses to the following question: “How difficult have these problems (related to anxiety) made it for you to get along with other people?” Source: Mozambique Impact Monitoring First Follow-up Survey (2021). Notes: A non-parametric Wilcoxon rank-sum test of the null hypothesis that for randomly selected values of each population (refugees and hosts), the probability of the value for refugees being greater than the value for hosts is equal to the probability of the value for hosts being greater than the value for refugees yields the following *p*-values: *p* = 0.33.

**Figure 5 healthcare-13-03187-f005:**
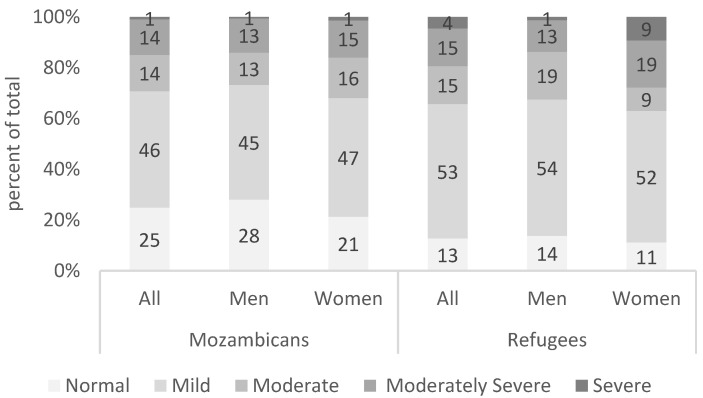
Prevalence of symptoms of depression. Source: Mozambique Impact Monitoring First Follow-up Survey (2021). Notes: A non-parametric Wilcoxon rank-sum test of the null hypothesis that for randomly selected values of each population (refugees and hosts), the probability of the value for refugees being greater than the value for hosts is equal to the probability of the value for hosts being greater than the value for refugees yields the following *p*-values: *p* = 0.02 (refugee versus hosts), *p* = 0.06 (men refugee vs. men hosts) and *p* = 0.08 (women refugee vs. women host).

**Figure 6 healthcare-13-03187-f006:**
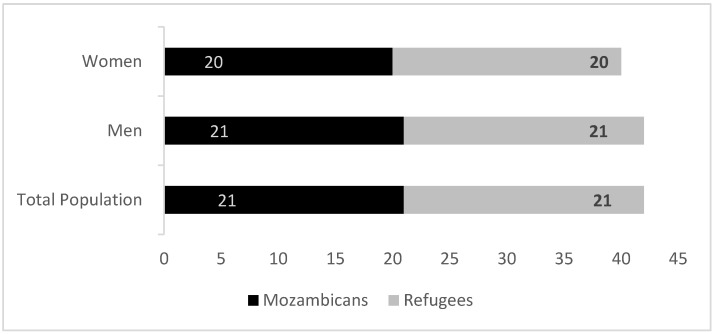
Average self-esteem index scores (out of 27) by gender. Source: Mozambique Impact Monitoring First Follow-up Survey (2021).

**Figure 7 healthcare-13-03187-f007:**
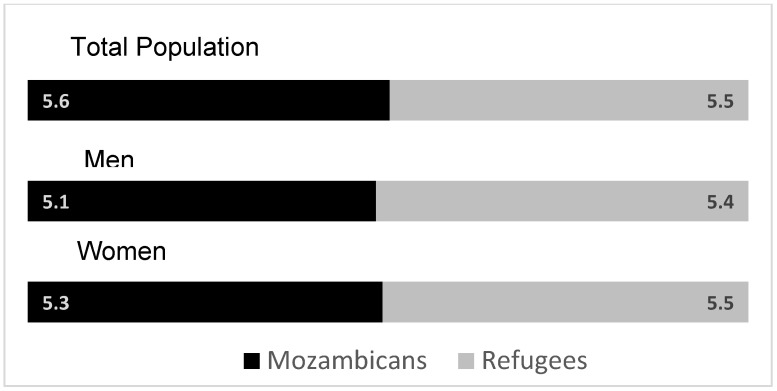
Average loneliness index scores (out of 10) by gender. Source: Mozambique Impact Monitoring First Follow-up Survey (2021). Notes: A non-parametric Wilcoxon rank-sum test of the null hypothesis that for randomly selected values of each population (refugees and hosts); the probability of the value for refugees being greater than the value for hosts is equal to the probability of the value for hosts being greater than the value for refugees yields the following *p*-values: *p* = 0.58 (refugees compared to Mozambicans), *p* = 0.15 (men refugees compared to Mozambican men) and *p* = 0.54 (women refugees compared to Mozambican women).

**Figure 8 healthcare-13-03187-f008:**
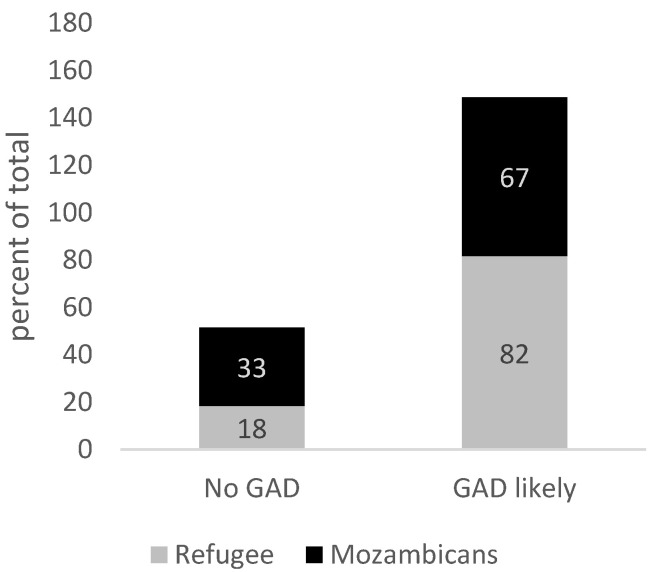
“Chronically Pessimistic” and incidence of anxiety. Source: Mozambique Impact Monitoring First Follow-up Survey (2021); Notes: A non-parametric Wilcoxon rank-sum test of the null hypothesis that for randomly selected values of each population (refugees and hosts), the probability of that value for refugees being greater than the value for hosts is equal to the probability of the value for hosts being greater than the value for refugees yields the following *p*-values: *p* = 0.06.

**Figure 9 healthcare-13-03187-f009:**
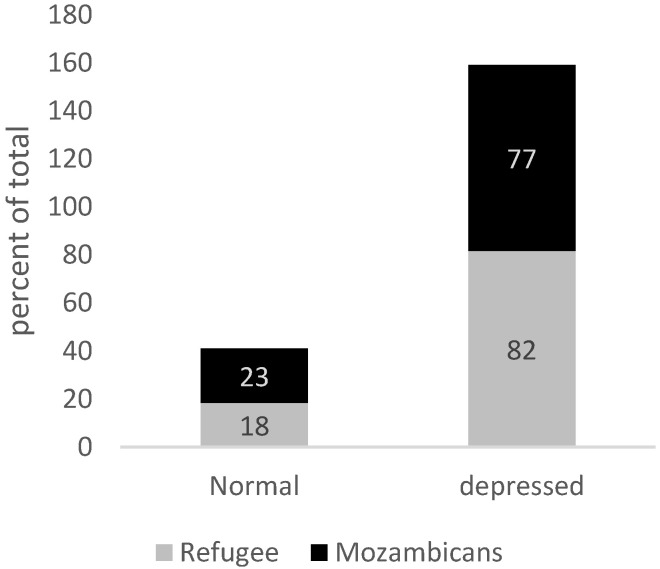
“Chronically Pessimistic” and incidence of depression. Source: Mozambique Impact Monitoring First Follow-up Survey (2021); Notes: A non-parametric Wilcoxon rank-sum test of the null hypothesis that for randomly selected values of each population (refugees and hosts), the probability of that value for refugees being greater than the value for hosts is equal to the probability of the value for hosts being greater than the value for refugees yields the following *p*-values: *p* = 0.00.

**Table 1 healthcare-13-03187-t001:** Descriptive characteristics across refugees and host community members.

	Refugees	Mozambicans	Difference
Percent of Male headed households	60%	52%	8%
Average age	39 years	36 years	2.7 **
Average household size	4.9	5.3	−0.4
Education Level			
No education	5%	25%	20% ***
Primary	27%	64%	−38% ***
Secondary	59%	10%	49% ***
Higher	7%	0%	7% ***
Marital Status			
Married/Domestic Partnership	47%	76%	−29% ***
Divorced/Separated/Widowed	25%	13%	12% **
Single/Never married	27%	8%	19% ***

Note: *, **, and *** indicate statistical significance at the 90%, 95%, and 99%, confidence levels, respectively. Source: Authors’ own calculation from Mozambique Impact Monitoring Wave 2 Survey (2021).

**Table 2 healthcare-13-03187-t002:** Prevalence rates of GAD and depression by self-esteem index quartiles.

Prevalence Rate	GAD	Depression
Total Population	25%	31%
High Self Esteem (top quartile)	3%	4%
Low Self-Esteem (bottom quartile)	65%	66%

Note: Because the population averages are not different between refugees and hosts, the analysis here is not disaggregated by population group (e.g., refugees and hosts).

**Table 3 healthcare-13-03187-t003:** Prevalence rates of GAD and depression by loneliness index quartiles.

Prevalence Rate	GAD	Depression
Total Population	25%	31%
Low Loneliness Score (bottom quartile)	32%	27%
High Loneliness Score (top quartile)	25%	29%

Note: Because the population averages are not different between refugees and hosts, the analysis here is not disaggregated by population group (e.g., refugees and hosts).

**Table 4 healthcare-13-03187-t004:** Depression, anxiety, financial security and socioeconomic characteristics.

	Individual Is Likely to Suffer from Generalized Anxiety Disorder (GAD)			Individual Is Likely to Suffer from Depression		
	Pooled Sample	Refugees	Mozambicans	Pooled Sample	Refugees	Mozambicans
	b/se	b/se	b/se	b/se	b/se	b/se
Financial Security Index	−0.069 **	0.017	−0.114 ***	−0.069 ***	−0.038	−0.088 **
	(0.030)	(0.049)	(0.037)	(0.026)	(0.036)	(0.034)
Age	0.000	−0.000	0.002	−0.001	0.001	−0.002
	(0.002)	(0.003)	(0.003)	(0.002)	(0.003)	(0.002)
Gender (1 = Women)	−0.061	−0.084	−0.040	−0.075 *	−0.180 **	−0.046
	(0.050)	(0.097)	(0.058)	(0.043)	(0.071)	(0.053)
Years of education	−0.005	−0.021	0.014	−0.019 **	−0.004	−0.034 **
	(0.011)	(0.015)	(0.017)	(0.009)	(0.011)	(0.015)
HH size	0.009	0.012	0.005	−0.003	0.009	−0.018
	(0.009)	(0.014)	(0.013)	(0.008)	(0.010)	(0.012)
Housing	−0.017	−0.000	−0.028	0.037 *	−0.001	0.071 **
	(0.022)	(0.033)	(0.031)	(0.019)	(0.024)	(0.029)
Speaks Portuguese	−0.034	−0.125	−0.032	0.050	0.195 ***	0.022
	(0.049)	(0.092)	(0.060)	(0.043)	(0.067)	(0.055)
Refugee (1 = Yes/0 = No)	0.037			0.168 ***		
	(0.072)			(0.062)		
Years in Mozambique (Refugees only)		−0.010			−0.001	
		(0.009)			(0.007)	
Distance to the centre of Maratane Settlement			−0.053 ***			0.005
			(0.019)			(0.017)
Constant	0.460 ***	0.734 ***	0.449 ***	0.765 ***	0.786 ***	0.904 ***
	(0.096)	(0.196)	(0.122)	(0.083)	(0.143)	(0.112)
r2	0.024	0.071	0.059	0.062	0.134	0.069
N	448	134	297	448	134	297

Note: *, ** and *** indicate statistical significance at 10%, 5% and 1%, respectively, and Monthly pay is converted from Meticals to USD at a rate of 1 Meticals = 0.016 USD. This analysis comes from the baseline survey. Sample size for host community reduced from 314 to 297 individuals because 17 respondents lacked accurate GPS coordinates. When including the control variable ‘Distance to the centre of Maratane Settlement’, these 17 households were excluded from the analysis.

## Data Availability

The original contributions presented in this study are included in the article. Further inquiries can be directed to the corresponding author. The data set and survey questionnaire are publicly available on the UNHCR Microdata Library here—Mozambique—Baseline Survey for Impact Evaluation of the UNHCR Implemented Graduation Program—2019: https://microdata.unhcr.org/index.php/catalog/1422, accessed on 10 August 2025.

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
