# Peer review of "Interconnected Challenges: Examining the Impact of Poverty, Disability, and Mental Health on Refugees and Host Communities in Northern Mozambique"

_healthcare, 2025, doi:10.3390/healthcare13243187_

Round 1
Reviewer 1 Report
Comments and Suggestions for Authors
This article provides an important and timely contribution to the literature on forced displacement by investigating the interconnections between poverty, disability, and mental health among both refugees and host communities in northern Mozambique. The comparative design is a major strength, as it captures dynamics that are often overlooked when refugee populations are studied in isolation. The use of validated instruments (PHQ-9, GAD-7, Rosenberg Self-Esteem, WG-SS) enhances methodological rigour, while the inclusion of less commonly analysed dimensions such as pessimism bias and self-esteem is innovative and adds depth to the analysis. The findings—particularly the high prevalence of depression and anxiety, the disproportionate burden on women, and the heightened vulnerabilities faced by persons with disabilities—are not only academically significant but also of direct policy relevance, highlighting the need for integrated mental health and livelihood support.
At the same time, some areas require further strengthening. The relatively small refugee sample size limits statistical power and makes it difficult to explore subgroup variations, while reliance on self-reported mental health screening tools introduces measurement noise compared to clinical assessments. Claims of causality should be framed more cautiously, as the study is primarily cross-sectional in nature. The discussion could also be enriched by deeper engagement with regional African mental health and displacement scholarship to situate the findings more firmly in their context. Finally, the policy section would benefit from more specific and actionable recommendations, particularly on how disability-inclusive livelihood and community-based psychosocial interventions could be operationalised in low-resource settings. Despite these limitations, the article is rigorous, innovative, and highly relevant, making a strong contribution to both academic and policy debates.
https://www.taylorfrancis.com/chapters/edit/10.4324/9781003567745-5/journeying-belong-divjyot-kaur-diotima-chattoraj-shuting-wong
Kindly refer to this book-chapter which highlights how factors such as openness, cultural intelligence, social support, and a sense of belonging facilitate positive adaptation and overall well-being, which are closely tied to mental health. While these women largely reported positive experiences and feelings of being “insiders,” the situation is very different for refugees, who often face forced displacement, legal and social barriers, trauma, and lack of access to supportive networks. These gaps significantly heighten acculturative stress and increase risks of anxiety, depression, and other mental health challenges. Thus, the findings underscore how access to protective factors can foster resilience, while their absence in refugee contexts directly undermines mental health outcomes.
Author Response
Comments and Suggestions for Authors
- This article provides an important and timely contribution to the literature on forced displacement by investigating the interconnections between poverty, disability, and mental health among both refugees and host communities in northern Mozambique. The comparative design is a major strength, as it captures dynamics that are often overlooked when refugee populations are studied in isolation. The use of validated instruments (PHQ-9, GAD-7, Rosenberg Self-Esteem, WG-SS) enhances methodological rigour, while the inclusion of less commonly analysed dimensions such as pessimism bias and self-esteem is innovative and adds depth to the analysis. The findings—particularly the high prevalence of depression and anxiety, the disproportionate burden on women, and the heightened vulnerabilities faced by persons with disabilities—are not only academically significant but also of direct policy relevance, highlighting the need for integrated mental health and livelihood support.
Response: Thank you for your thoughtful and encouraging feedback. We greatly appreciate your recognition of the comparative design, methodological rigor, and the inclusion of innovative dimensions such as pessimism bias and self-esteem. We are pleased that you found the findings both academically significant and policy relevant, and we hope the paper contributes meaningfully to ongoing discussions around integrated support for displaced populations.
- At the same time, some areas require further strengthening. The relatively small refugee sample size limits statistical power and makes it difficult to explore subgroup variations, while reliance on self-reported mental health screening tools introduces measurement noise compared to clinical assessments. Claims of causality should be framed more cautiously, as the study is primarily cross-sectional in nature.
Response: Thank you for the important observation around the statistical limits of a relatively small sample size. We acknowledge that the relatively small refugee sample size limits statistical power and constrains our ability to explore subgroup heterogeneity in greater depth. We have added a note in the limitations section to clarify this point and updated the paper throughout accordingly to ensure this comment is accurately reflected. Regarding the use of self-reported mental health screening tools, we agree that these instruments may introduce measurement noise compared to clinical assessments. However, given the logistical and ethical constraints of conducting clinical evaluations in humanitarian settings, validated self-reported tools such as the PHQ-9 and GAD-7 remain widely accepted and feasible alternatives. That said two of the authors in this paper have ongoing work in Bangladesh to validate self-reported tools with clinical evaluations, so we could not agree with you more on the importance. We have added a brief discussion of this trade-off in the methods and limitations sections to provide additional context and have added this point in our conclusion to stress the need for additional work that validates self-reported mental health tools such as the PHQ-9 and GAD-7.
- The discussion could also be enriched by deeper engagement with regional African mental health and displacement scholarship to situate the findings more firmly in their context. Finally, the policy section would benefit from more specific and actionable recommendations, particularly on how disability-inclusive livelihood and community-based psychosocial interventions could be operationalised in low-resource settings. Despite these limitations, the article is rigorous, innovative, and highly relevant, making a strong contribution to both academic and policy debates.
Response: Thank you for this valuable suggestion. We agree that deeper engagement with regional African scholarship on mental health and displacement would strengthen the contextual grounding of our discussion. In response, we have expanded the discussion section to incorporate relevant literature from the region, including recent studies on psychosocial wellbeing in displacement settings across sub-Saharan Africa. For example, we now reference a comprehensive review paper by Gbadamosi et al. (2022) on depression in Sub-Saharan Africa and the studies from Mozambique such as those by Audet et al (2018) on depression among female heads-of-household in rural Mozambique and by Antabe et al (2025) reporting on depression and anxiety in the Mozambique Demographic and Health Survey 2022.
We also appreciate your recommendation to make the policy section more specific and actionable. We have revised this section to include concrete examples of how disability-inclusive livelihood and community-based psychosocial interventions could be operationalized in low-resource settings, drawing on existing programmatic models and implementation frameworks from similar contexts.
We are grateful for your recognition of the article’s rigor and relevance, and for your constructive feedback which has helped us improve the manuscript.
- https://www.taylorfrancis.com/chapters/edit/10.4324/9781003567745-5/journeying-belong-divjyot-kaur-diotima-chattoraj-shuting-wong: Kindly refer to this book-chapter which highlights how factors such as openness, cultural intelligence, social support, and a sense of belonging facilitate positive adaptation and overall well-being, which are closely tied to mental health. While these women largely reported positive experiences and feelings of being “insiders,” the situation is very different for refugees, who often face forced displacement, legal and social barriers, trauma, and lack of access to supportive networks. These gaps significantly heighten acculturative stress and increase risks of anxiety, depression, and other mental health challenges. Thus, the findings underscore how access to protective factors can foster resilience, while their absence in refugee contexts directly undermines mental health outcomes.
Response: Thank you for the suggested reference. We agree that the chapter by Kaur, Chattoraj, and Wong offers valuable insights into the role of protective factors such as social support and cultural intelligence in facilitating well-being. We also agree with you that the focus of the chapter on privileged Asian migrant women in Singapore presents a context that differs significantly from that of refugees. Refugees often experience forced displacement, legal and social exclusion, and trauma, which fundamentally shape their mental health outcomes and adaptation processes. As such, the mechanisms discussed in the chapter may not be directly applicable to refugee populations, whose experiences are shaped by distinct structural and psychosocial challenges. Given space constraints, we have not included this citation, prioritizing literature most directly relevant to our context and results.
Reviewer 2 Report
Comments and Suggestions for Authors
This manuscript addresses a critical intersection of public health, forced displacement, and social determinants of health. The direct, data-driven comparison of mental health and disability burdens between impoverished refugee and host communities in Northern Mozambique is a dataset that is exceptionally rare and valuable, and the use of well-validated instruments like the PHQ-9, GAD-7, and the Washington Group Short Set on Functioning lends significant credibility to the findings. The study’s focus on a population that is often invisible in the literature is a major contribution. I was also impressed by the innovative inclusion of psychosocial factors like self-esteem and pessimism, which provides a more nuanced understanding of mental health predictors beyond typical socioeconomic variables.
While there is much to like about this manuscript, there are several concerns that I have:
- The manuscript lacks an explicit theoretical framework. While the introduction does an effective job of reviewing relevant literature, it stops short of grounding the study in a specific theory (e.g., an intersectional framework, the stress-process model) that could guide the selection of variables and interpretation of their interplay. An explicit framework would provide a stronger theoretical scaffold, helping to explain the mechanisms behind the observed associations between poverty, disability, displacement status, and mental health.
- My most significant concern is the repeated claim of establishing causality. The manuscript’s design is cross-sectional, and the statistical methods used (e.g., OLS regression) can only establish association or correlation. The cross-sectional approach makes it impossible to determine the directionality of the observed relationships; for instance, does financial insecurity lead to depression, or does depression impair one’s ability to work and thus lead to financial insecurity? This is a classic limitation that cannot be overcome with this data. However, the authors state they are examining a causal relationship in both the introduction and conclusion. This is a fundamental, erroneous overstatement of the findings. All language suggesting causality should be revised to reflect the associative nature of the research, e.g., “relationship”, “association”, “correlation”, etc.
- There are a few points in the methodology that require clarification and should be more clearly addressed as limitations:
- The sampling strategy, which relies on a poverty scorecard to identify “ultra-poor” households, is a form of purposive sampling. While appropriate for the study’s aim, this creates a significant selection bias that severely limits the generalizability of the findings to the broader refugee and host populations. The authors should be more explicit that their results reflect the conditions of the most impoverished segment of these communities, not the communities as a whole.
- Additionally, the authors note that for participants not fluent in Portuguese, survey questions were verbally translated into Emakua or Swahili by enumerators. While the enumerators were proficient, this on-the-spot translation, rather than the use of a formally validated and back-translated survey instrument in these languages, could introduce measurement error, as cultural nuances critical to expressing mental distress may be lost.
- The measure for “chronically pessimistic”, defined as ranking one’s life a 3 or below currently and in five years, appears to be a novel, author-defined construct. While the finding is interesting, the manuscript would be strengthened by either citing a precedent for this operationalization or explicitly acknowledging its novelty and providing a stronger justification for the chosen cut-off.
- The authors acknowledge the limitation of using a four-question version of the UCLA Loneliness Scale. The discussion could go further, however. Given the surprising null finding for loneliness as a predictor of depression or anxiety, it would be useful for the authors to offer ideas on why this might be. Is it possible the shortened scale failed to capture the construct adequately in this specific cultural context, or could it be that in such a communally challenging environment, loneliness manifests differently than in the Western contexts where the scale was developed?
- The interpretation of some findings could be deepened. The study treats “refugees” and “hosts” as two monolithic categories for comparison. However, the refugee population is noted to be diverse, with individuals from the DRC, Burundi, Rwanda, and Somalia, while the host community is primarily Makua. This approach risks conceptual homogenization, potentially obscuring critical differences in traumatic exposures, cultural backgrounds, and integration experiences that exist within the refugee group. Acknowledging this as a limitation would add important context. For instance, the result that financial insecurity is significantly associated with mental health issues for the host community but not for refugees is fascinating. The authors suggest this may be due to the smaller refugee sample size, which is a valid point. However, are there other potential explanations worth exploring? The descriptive data show refugees have significantly higher levels of secondary and higher education. Could education, or other unmeasured factors like social cohesion within specific national subgroups of the refugee community, serve as a buffer against the mental health impacts of financial insecurity? A more speculative paragraph exploring these alternative explanations would enrich the discussion.
- Finally, there are a few minor but important points of clarity and accuracy that should be addressed during revision. From a structural and stylistic standpoint, the introduction refers to the current study (e.g., “We also innovate by looking at…” and “This study adds to the literature by looking at a forced displacement context …) before the methods section. Consistent with academic convention, the introduction should focus exclusively on the existing literature and the knowledge gap, with the only reference to the present study being the statement of aims or research questions at the very end. The paragraph beginning “We also innovate…” is also awkwardly placed and disrupts the flow of the literature review.
A thorough proofread is also needed to catch several typos and grammatical issues. For example:
- The abstract currently stands at ~ 443 words, which is substantially longer than the ~250-word limit outlined in Healthcare’s instruction for authors. It should be edited for brevity and conciseness to adhere to the journal’s guidelines.
- On page 3, there is a duplicated word: “…host communities in in a forced displaced setting…”.
- On page 7, the phrasing “…though results not significantly significant” should be corrected to “…though the results are not statistically significant.”
- On page 9, there appears to be a stray number in the text: “…29 percent hosts (5)”, which should likely be a reference to Figure 5.
- In Table 4, the currency exchange rate in the note, “1 Meticals = 0.16 USD”, appears to be a typo and should likely be 0.016 USD.
- The citation style is inconsistent, switching between bracketed numbers (e.g., “[26], [27]”) and parenthetical numbers (e.g., “(19,35)” ). This should be standardized according to the journal’s guidelines.
- The title of Figure 9 is “Chronically Depressed and Incidence of Depression”, but the text clearly identifies the variable as “chronically pessimistic”; this must be corrected.
- The sample sizes in the regression analyses in Table 4 are inconsistent. The text states the total sample is 448, but the pooled regressions show N=448 for depression and N=297 for anxiety. The reason for this discrepancy should be explicitly stated.
- There are also several references to future dates (e.g., data availability in 2026, refugee statistics as of January 2025), which should be revised to reflect the time of writing.
In sum, while this paper has the potential to be an excellent and impactful contribution to the literature on refugee and host community health, significant revisions are required. Most importantly, the adoption of an explicit theoretical framework and the reframing of the entire manuscript to accurately represent the findings as correlational rather than causal is imperative. Additional clarifications regarding the limitations of the sampling and translation methodology, the conceptual homogenization of the study groups, the novel pessimism measure, the null finding for loneliness, the interpretation of the financial security results, and the correction of the aforementioned stylistic and grammatical issues are also necessary to strengthen this work
Author Response
- This manuscript addresses a critical intersection of public health, forced displacement, and social determinants of health. The direct, data-driven comparison of mental health and disability burdens between impoverished refugee and host communities in Northern Mozambique is a dataset that is exceptionally rare and valuable, and the use of well-validated instruments like the PHQ-9, GAD-7, and the Washington Group Short Set on Functioning lends significant credibility to the findings. The study’s focus on a population that is often invisible in the literature is a major contribution. I was also impressed by the innovative inclusion of psychosocial factors like self-esteem and pessimism, which provides a more nuanced understanding of mental health predictors beyond typical socioeconomic variables.
Response: Thank you very much for your thoughtful and encouraging feedback. We deeply appreciate your recognition of the study’s contribution to understanding the intersection of public health, forced displacement, and social determinants of health. We agree that the rarity of comparable data between refugee and host populations in such settings makes this analysis particularly valuable, and we are pleased that the use of validated instruments such as the PHQ-9, GAD-7, and the Washington Group Short Set on Functioning helped strengthen the credibility of our findings.
We are especially grateful for your remarks on the inclusion of psychosocial factors like self-esteem and pessimism. Our intention was to move beyond conventional socioeconomic indicators and capture a more holistic picture of mental health vulnerabilities in these communities. Your recognition of this approach is very encouraging.
- While there is much to like about this manuscript, there are several concerns that I have:
- The manuscript lacks an explicit theoretical framework. While the introduction does an effective job of reviewing relevant literature, it stops short of grounding the study in a specific theory (e.g., an intersectional framework, the stress-process model) that could guide the selection of variables and interpretation of their interplay. An explicit framework would provide a stronger theoretical scaffold, helping to explain the mechanisms behind the observed associations between poverty, disability, displacement status, and mental health.
Response: Thank you for this thoughtful suggestion. Our goal in this paper is deliberately descriptive: to provide comparable, within-setting evidence on poverty, disability, and mental health among ultra-poor refugees and hosts. Because our cross-sectional design cannot adjudicate mechanisms, anchoring the study in any single theory would risk over-interpreting associations. We have clarified in the Introduction and Methods that our contribution is an empirical mapping to inform future, theory-driven causal work, and we avoid causal language accordingly.
- My most significant concern is the repeated claim of establishing causality. The manuscript’s design is cross-sectional, and the statistical methods used (e.g., OLS regression) can only establish association or correlation. The cross-sectional approach makes it impossible to determine the directionality of the observed relationships; for instance, does financial insecurity lead to depression, or does depression impair one’s ability to work and thus lead to financial insecurity? This is a classic limitation that cannot be overcome with this data. However, the authors state they are examining a causal relationship in both the introduction and conclusion. This is a fundamental, erroneous overstatement of the findings. All language suggesting causality should be revised to reflect the associative nature of the research, e.g., “relationship”, “association”, “correlation”, etc.
Response: We completely agree that the study is primarily descriptive. We have revised the manuscript to state explicitly that our analysis is descriptive and does not advance causal claims.
- There are a few points in the methodology that require clarification and should be more clearly addressed as limitations:
- The sampling strategy, which relies on a poverty scorecard to identify “ultra-poor” households, is a form of purposive sampling. While appropriate for the study’s aim, this creates a significant selection bias that severely limits the generalizability of the findings to the broader refugee and host populations. The authors should be more explicit that their results reflect the conditions of the most impoverished segment of these communities, not the communities as a whole.
Response: Thank you for this helpful point. We fully agree. In the revised manuscript, we have stated more prominently that findings pertain to the most impoverished segment rather than the full refugee/host populations. We will also adjust language throughout to avoid population-wide inferences.
- Additionally, the authors note that for participants not fluent in Portuguese, survey questions were verbally translated into Emakua or Swahili by enumerators. While the enumerators were proficient, this on-the-spot translation, rather than the use of a formally validated and back-translated survey instrument in these languages, could introduce measurement error, as cultural nuances critical to expressing mental distress may be lost.
Response: Thank you, we agree that this is a real limitation we cannot fully remedy ex post. In the revised version of the manuscript in the limitations section we have noted that although the enumerators were bilingual and trained, we did not deploy formally translated/back-translated Emakua/Swahili instruments, which may introduce measurement error and cultural misalignment in mental-health items. Instead, we extensively took several mitigation steps such as using standardized glossaries during training, role-plays, ensuring that all enumerators were fluent in both Portuguese and the local language, etc.
The measure for “chronically pessimistic”, defined as ranking one’s life a 3 or below currently and in five years, appears to be a novel, author-defined construct. While the finding is interesting, the manuscript would be strengthened by either citing a precedent for this operationalization or explicitly acknowledging its novelty and providing a stronger justification for the chosen cut-off.
The authors acknowledge the limitation of using a four-question version of the UCLA Loneliness Scale. The discussion could go further, however. Given the surprising null finding for loneliness as a predictor of depression or anxiety, it would be useful for the authors to offer ideas on why this might be. Is it possible the shortened scale failed to capture the construct adequately in this specific cultural context, or could it be that in such a communally challenging environment, loneliness manifests differently than in the Western contexts where the scale was developed?
Response: Thank you for raising this point. We now expand the discussion of the loneliness null to consider two possibilities: (i) that it results from attenuation from using a shortened 4-item index, and (ii) that it is a context-specific manifestation of social connectedness in Maratane/Nampula that may not map cleanly onto standard loneliness items. [We also add brief robustness checks (reported in an appendix): internal consistency of the 4 items, item-level associations with PHQ-9/GAD-7, alternative index constructions, and simple heterogeneity by refugee/host and gender. ]
We include the following text in the manuscript:
“The lack of association between loneliness and depression/anxiety should be interpreted cautiously. First, using a 4-item version of the UCLA scale likely increases measurement error, biasing estimates toward zero. Second, in this setting social strain may manifest through obligations, crowding, or network dependence rather than perceived isolation, which our items may capture imperfectly. ”
- The interpretation of some findings could be deepened. The study treats “refugees” and “hosts” as two monolithic categories for comparison. However, the refugee population is noted to be diverse, with individuals from the DRC, Burundi, Rwanda, and Somalia, while the host community is primarily Makua. This approach risks conceptual homogenization, potentially obscuring critical differences in traumatic exposures, cultural backgrounds, and integration experiences that exist within the refugee group. Acknowledging this as a limitation would add important context. For instance, the result that financial insecurity is significantly associated with mental health issues for the host community but not for refugees is fascinating. The authors suggest this may be due to the smaller refugee sample size, which is a valid point. However, are there other potential explanations worth exploring? The descriptive data show refugees have significantly higher levels of secondary and higher education. Could education, or other unmeasured factors like social cohesion within specific national subgroups of the refugee community, serve as a buffer against the mental health impacts of financial insecurity? A more speculative paragraph exploring these alternative explanations would enrich the discussion.
Response: Thank you, we agree with this point. We now add language flagging that our “refugee” and “host” categories can aggregate substantial within-group heterogeneity (national origin, migration histories, social networks, time in Mozambique, and exposure to trauma), and that our sample is underpowered for meaningful subgroup analysis. We also expand the discussion with a brief, clearly speculative paragraph outlining plausible mechanisms such as education differences, subgroup social cohesion, differential access to assistance, and reporting norms, among others. This could attenuate the observed association between financial insecurity and mental health among refugees. We emphasize these as hypotheses for future, adequately powered work.
We add the following paragraph to our discussion: “Our refugee and host categories can potentially mask within-group heterogeneity. Among refugees, national-origin subgroups differ in exposure to trauma, time since displacement, language, and network structure, all of which may shape mental health and its relationship to economic stress. In our data, refugees also have higher average schooling than hosts. Education, stronger enclave networks, differential access to NGO support, or stigma/reporting norms could plausibly buffer the mental-health impact of short-run financial insecurity, yielding smaller observable associations even absent true differences. Given sample size constraints, we are underpowered to test these mechanisms or to present credible subgroup estimates, and we avoid over-interpreting pooled contrasts. We flag targeted, adequately powered heterogeneity analyses as a priority for future work.”
- Finally, there are a few minor but important points of clarity and accuracy that should be addressed during revision. From a structural and stylistic standpoint, the introduction refers to the current study (e.g., “We also innovate by looking at…” and “This study adds to the literature by looking at a forced displacement context …) before the methods section. Consistent with academic convention, the introduction should focus exclusively on the existing literature and the knowledge gap, with the only reference to the present study being the statement of aims or research questions at the very end. The paragraph beginning “We also innovate…” is also awkwardly placed and disrupts the flow of the literature review.
Response: Thanks for highlighting this point. We are a mixed researcher group of economists and mental health specialists. In response to this point, we have edited the introduction to align better with the convention of biomedical journals. In the introduction, we now more strictly to focus on the existing literature and the knowledge gap and deleted descriptions of the current research, except for the last part where we describe which issues we wish to explore in the paper, making a segway to the Methods section.
A thorough proofread is also needed to catch several typos and grammatical issues. For example:
- The abstract currently stands at ~ 443 words, which is substantially longer than the ~250-word limit outlined in Healthcare’s instruction for authors. It should be edited for brevity and conciseness to adhere to the journal’s guidelines.
Response: The revised manuscript has an abstract that is now within the 250-world limit.
- On page 3, there is a duplicated word: “…host communities in in a forced displaced setting…”.
Response: Thanks for catching this typo. It has now been removed.
- On page 7, the phrasing “…though results not significantly significant” should be corrected to “…though the results are not statistically significant.”
Response: Thanks for catching this typo. It has now been fixed.
- On page 9, there appears to be a stray number in the text: “…29 percent hosts (5)”, which should likely be a reference to Figure 5.
Response: Thanks for catching this typo. It has now been corrected.
- In Table 4, the currency exchange rate in the note, “1 Meticals = 0.16 USD”, appears to be a typo and should likely be 0.016 USD.
Response: Thanks for catching this typo. It has now been fixed.
- The citation style is inconsistent, switching between bracketed numbers (e.g., “[26], [27]”) and parenthetical numbers (e.g., “(19,35)” ). This should be standardized according to the journal’s guidelines.
Response: Thanks for catching this typo. It has now been corrected.
- The title of Figure 9 is “Chronically Depressed and Incidence of Depression”, but the text clearly identifies the variable as “chronically pessimistic”; this must be corrected.
Response: Thanks for catching this typo. It has now been changed.
- The sample sizes in the regression analyses in Table 4 are inconsistent. The text states the total sample is 448, but the pooled regressions show N=448 for depression and N=297 for anxiety. The reason for this discrepancy should be explicitly stated.
Response: Note that in Table 4, all indicators have the same number of observations. The pool sample consists of 448 observations while the Mozambican host community sample corresponds to 297 observations.
- There are also several references to future dates (e.g., data availability in 2026, refugee statistics as of January 2025), which should be revised to reflect the time of writing.
Response: Thanks for raising this point. We have now corrected these issues.
In sum, while this paper has the potential to be an excellent and impactful contribution to the literature on refugee and host community health, significant revisions are required. Most importantly, the adoption of an explicit theoretical framework and the reframing of the entire manuscript to accurately represent the findings as correlational rather than causal is imperative. Additional clarifications regarding the limitations of the sampling and translation methodology, the conceptual homogenization of the study groups, the novel pessimism measure, the null finding for loneliness, the interpretation of the financial security results, and the correction of the aforementioned stylistic and grammatical issues are also necessary to strengthen this work
Response: Thanks again for these excellent points and suggestions. We have now revised the manuscript addressing the issues you raise. We hope you agree with us that the paper is now significantly stronger.
Round 2
Reviewer 2 Report
Comments and Suggestions for Authors
The authors have successfully addressed most of my major concerns – particularly the critical issues related to causality and interpretation. The manuscript is therefore significantly stronger. However, they have failed to address two of my methodological points (one of which they ignored entirely):
- The authors did not respond to the pessimism measure. The text still presents the "chronically pessimistic" measure (ranking life ≤ 3 now and in five years) as a given, without citing a precedent, acknowledging its novelty, or providing a justification for the cut-off.
- The authors' response to my sample size concern feels dismissive and failed to identify the real issue. The methods section states the sample includes 134 refugees and 314 Mozambican hosts, for a total N of 448. However, Table 4 clearly shows the regressions for the Mozambican subgroup were run on an N of 297. The authors never explain why 17 Mozambican participants are missing from the regression analysis. Such attrition is a methodological detail that must be addressed.
Author Response
Comments and Suggestions for Authors Second R&R
- The authors have successfully addressed most of my major concerns – particularly the critical issues related to causality and interpretation. The manuscript is therefore significantly stronger. However, they have failed to address two of my methodological points (one of which they ignored entirely):
- The authors did not respond to the pessimism measure. The text still presents the "chronically pessimistic" measure (ranking life ≤ 3 now and in five years) as a given, without citing a precedent, acknowledging its novelty, or providing a justification for the cut-off.
RESPONSE: We appreciate this important point and apologize for not fully addressing it to the reviewers satisfaction in the previous version. We have already cited in the literature the established link between depression and pessimism bias on page 5 of the current manuscript: “In particular, dispositional optimism and pessimism are noted as key predictors of depression risk in numerous studies [35], [36], [37], [38], as these traits are closely linked to the absence of positive emotions, a core symptom of depression [39].” After reviewing the reviewers previous comments from the first R&R including: “The measure for “chronically pessimistic”, defined as ranking one’s life a 3 or below currently and in five years, appears to be a novel, author-defined construct. While the finding is interesting, the manuscript would be strengthened by either citing a precedent for this operationalization or explicitly acknowledging its novelty and providing a stronger justification for the chosen cut-off,” we have added the following text in the paper in the limitations section: “Further, the chronically pessimistic scale is the author’s invention and the cut-off of a score of a 3 out of 10 for both today and in five years would benefit from more robust experimentation to validate the cut-offs selected.”
- The authors' response to my sample size concern feels dismissive and failed to identify the real issue. The methods section states the sample includes 134 refugees and 314 Mozambican hosts, for a total N of 448. However, Table 4 clearly shows the regressions for the Mozambican subgroup were run on an N of 297. The authors never explain why 17 Mozambican participants are missing from the regression analysis. Such attrition is a methodological detail that must be addressed.
RESPONSE: Thank you for this important observation and apologies for somehow failing to detail correctly in our previous reply. We have now added the following note to the table to explain this important difference. “Sample size for host community reduced from 314 to 297 individuals because 17 respondents lacked accurate GPS coordinates. When including the control variable ‘Distance to the center of Maratane Settlement,’ these 17 households were excluded from the analysis.”